# Insights into Manganese Superoxide Dismutase and Human Diseases

**DOI:** 10.3390/ijms232415893

**Published:** 2022-12-14

**Authors:** Mengfan Liu, Xueyang Sun, Boya Chen, Rongchen Dai, Zhichao Xi, Hongxi Xu

**Affiliations:** 1School of Pharmacy, Shanghai University of Traditional Chinese Medicine, Shanghai 201203, China; 2Engineering Research Center, Shanghai Colleges for TCM New Drug Discovery, Shanghai 201203, China; 3Zhejiang Province Key Laboratory of Anti-Cancer Drug Research, Institute of Pharmacology and Toxicology, College of Pharmaceutical Sciences, Zhejiang University, Hangzhou 310058, China

**Keywords:** MnSOD, oxidative stress, ROS, human diseases, MnSOD mimetics

## Abstract

Redox equilibria and the modulation of redox signalling play crucial roles in physiological processes. Overproduction of reactive oxygen species (ROS) disrupts the body’s antioxidant defence, compromising redox homeostasis and increasing oxidative stress, leading to the development of several diseases. Manganese superoxide dismutase (MnSOD) is a principal antioxidant enzyme that protects cells from oxidative damage by converting superoxide anion radicals to hydrogen peroxide and oxygen in mitochondria. Systematic studies have demonstrated that MnSOD plays an indispensable role in multiple diseases. This review focuses on preclinical evidence that describes the mechanisms of MnSOD in diseases accompanied with an imbalanced redox status, including fibrotic diseases, inflammation, diabetes, vascular diseases, neurodegenerative diseases, and cancer. The potential therapeutic effects of MnSOD activators and MnSOD mimetics are also discussed. Targeting this specific superoxide anion radical scavenger may be a clinically beneficial strategy, and understanding the therapeutic role of MnSOD may provide a positive insight into preventing and treating related diseases.

## 1. Introduction

Reactive oxygen species (ROS) are a series of molecular oxygen derivatives that regulate numerous physiological and pathological processes [1]. As a group of chemical species, ROS can be categorized into non-radical and free radical species. Non-radical ROS include hydrogen peroxide (H_2_O_2_), organic hydroperoxides (ROOH), singlet molecular oxygen (^1^O_2_), ozone (O_3_), hypochlorous acid (HOCl), and hypobromous acid (HOBr), whereas free radical ROS include superoxide anion radical (O_2_^•−^), hydroxyl radical (·OH), peroxyl radical (ROO·) and alkoxyl radical (RO·) [2]. Cellular ROS are generated endogenously through mitochondrial oxidative phosphorylation, or exogenously stimulated by xenobiotics, cytokines, and bacterial infection [3,4]. O_2_^•−^ and H_2_O_2_ are acknowledged as the most physiologically relevant ROS. O_2_^•−^ is mainly generated by nicotinamide adenine dinucleotide phosphate oxidases (NOXs), or the complexes I and III of the mitochondrial electron transport chain (ETC). H_2_O_2_ is generated by superoxide dismutases (SOD), NOX4, monoamine oxidases, and xanthine oxidases within a few different organelles (endoplasmic reticulum, mitochondria, and peroxisomes). More importantly, ROS are required for numerous cellular processes including cell growth, differentiation, and death by acting as signalling molecules [5]. Multiple signalling pathways are involved in mediating ROS, including nuclear factor kappa-B (NF-κB), mitogen-activated protein kinase (MAPK) cascade, Kelch-like ECH-associated protein 1-nuclear factor erythroid 2-related factor 2-antioxidant response elements (Keap1-Nrf2-ARE) signalling, adenosine 5′-monophosphate-activated protein kinase (AMPK), phosphoinositide-3 kinase-(PI3K-) Akt pathway, etc. [6,7].

The dynamic balance between ROS production and antioxidant capacity responsibly maintains the cellular redox homeostasis [8]. However, oxidative stress occurs when elevated ROS overwhelms the cellular antioxidant defence, damaging nucleic acids, proteins, and lipids [4,5,9]. To protect cells from oxidative damage as well as to maintain physiological ROS levels, organisms have developed antioxidant defence systems that comprise small-molecular-weight antioxidants and antioxidant enzymes. Small-molecular-weight antioxidants include glutathione (GSH), cysteine, ascorbic acid, and α-tocopherol, while antioxidant enzymes include SOD, catalase (CAT), glutathione peroxidase (GPX), glutaredoxin (GRX), thioredoxin (TXN), and peroxiredoxin (PRX) [10,11,12]. From a pharmacokinetic point of view in scavenging intracellular ROS, targeting antioxidant enzymes is more effective than small-molecule antioxidants, and ultimately maintains the antioxidant defence [13]. A series of antioxidant enzymes such as CAT, GPX, and PRX eliminate H_2_O_2_ by converting H_2_O_2_ into H_2_O [14,15,16]. On the other hand, SOD is the most powerful O_2_^•−^ scavenger, which catalyses the dismutation of O_2_^•−^ into H_2_O_2_ and O_2_ [17]. Currently, three isoforms of SOD have been discovered in mammalian cells, namely, copper–zinc superoxide dismutase (Cu/ZnSOD, SOD1), manganese superoxide dismutase (MnSOD, SOD2), and the extracellular superoxide dismutase (EcSOD, SOD3) [5]. Of these isoforms, MnSOD garners widespread interest as it potently scavenges mitochondrial O_2_^•−^, which is the major source of cellular ROS [18]. The indispensable role of MnSOD is further highlighted by the evidence of significant neonatal mortality in MnSOD-deficient mice, compared to the considerable neonatal survival in the SOD1- or SOD3-deficient mice [19,20,21,22].

MnSOD is a nuclear-encoded enzyme that translocates into the mitochondrial matrix [12]. It is constituted by a homotetramer containing an active site with manganese as a cofactor. Despite its enzymatic antioxidant function, MnSOD readily binds with iron, and the produced iron-substituted enzyme attains peroxidase activity that generates highly reactive hydroxyl radicals [23,24]. Therefore, in protecting against oxidative stress, manganese seems like the perfect metal, while the quaternary structure of MnSOD responsibly maintains the catalytic and dismutase activity [25]. The expression and activity of MnSOD is regulatable at multiple levels, from transcription and translation to posttranslational modifications [26]. NF-κB, specificity protein 1 (Sp1), activating protein-1 (Ap1), p53, and CCAAT binding protein (C/EBP) are the major transcription factors that regulate MnSOD gene expression by directly binding to specific DNA elements or interacting with its partners [27,28]. Furthermore, posttranslational modifications regulate MnSOD protein expression and activity through nitration, phosphorylation, and acetylation. Peroxynitrite (ONOO^-^) inactivates MnSOD by nitrating the tyrosine-34 amino acid residue of MnSOD protein. Acetylation at lysine-68, lysine-122, lysine-53, and lysine-89 amino acid residues, also inactivate MnSOD. In addition, sirtuin 3 (SIRT3) reactivates MnSOD by deacetylating the above lysine residues in response to irradiation, nutritional deprivation, and oxidative stress [25,27]. Moreover, MnSOD activity and stability are enhanced through phosphorylation at Ser106 by the mitochondrial cell-cycle kinase 1 (Cdk1) [29].

Considering that MnSOD engages in detoxifying ROS through O_2_^•−^ dismutation, which possibly becomes a contributing factor or consequence in multiple diseases, it is vital to understand the physiological and pathological role of MnSOD. Recent evidence has revealed the pathological involvement of MnSOD in a number of diseases (Figure 1). For example, the enzymatic activity or expression of MnSOD is frequently downregulated in numerous diseases, such as diabetes and neurodegenerative diseases [30,31,32,33], whereas overexpression of MnSOD protects against pro-oxidant insults resulting from inflammatory cytokines, irradiation, hyperoxic injury, and ischaemia/reperfusion [34,35,36,37,38,39,40]. In addition, selective MnSOD mimetics have therapeutic potential in treating rheumatoid arthritis (RA), ischaemic stroke, and kidney diseases by mimicking the activity of MnSOD [41,42,43,44]. Nevertheless, a comprehensive summary of MnSOD regulatory roles in a range of diseases is lacking. In this review, we focus on the mechanisms of MnSOD in regulating various diseases that are associated with imbalanced redox status, including inflammation, fibrotic diseases, diabetes, neurodegenerative diseases, vascular diseases, and cancer (Figure 2). Advances in the therapeutic application of MnSOD activators and MnSOD mimetics are also discussed (Table 1). Moreover, limitations and critical issues in these MnSOD-related studies are addressed. Further understanding of how MnSOD can be regulated in the clinic may provide strategies and prospects for antioxidant therapy. Additionally, the comprehensive information and scientific advances presented in this review may facilitate the understanding of the physiological and pathological roles of MnSOD.

## 2. MnSOD in Diseases

### 2.1. Inflammatory Diseases

Inflammation acts as a defensive mechanism to confront harmful stimuli such as infection and tissue damage [64,65]. Oxidative stress in activated infiltrating immune cells and tissue-resident immune cells, such as macrophages and T cells, is responsible for inflammation-related diseases [66,67]. Numerous studies have demonstrated that MnSOD plays a vital role in protecting organismal integrity and homeostasis by regulating inflammatory chemokines and cytokines [68,69] (Figure 2).

#### 2.1.1. Pulmonary Inflammation

MnSOD was found to attenuate pulmonary inflammation in different models of lung damage [34,39,70,71]. Exposure to high oxygen concentrations can cause hyperoxic lung injury, characterized by the accumulation of toxic ROS and the production of proinflammatory cytokines in the lung, such as interleukin (IL)-6, IL-8, and IL-1β [16,71,72]. MnSOD elevation weakened hyperoxia-induced A549 cell growth inhibition, at least in part by suppressing ROS-induced IL-8 production [34]. Likewise, MnSOD overexpression in the lung epithelium of transgenic mice prevented hyperoxic lung injury and improved the survival rate after exposure to 95% O_2_ [39]. Targeting MnSOD is also a promising strategy for treating radiation-induced lung damage. Chen et al. revealed that the tail vein injection of MnSOD-overexpressing mesenchymal stem cells (MnSOD-MSCs) alleviated radiation-induced lung histopathological injury and improved the mouse survival rate. Mechanistically, MnSOD-MSCs significantly mitigated lung inflammation by impeding proinflammatory cytokine activation (tumour necrosis factor (TNF)-α, IL-1β, and IL-6) and strengthening the anti-inflammatory cytokine (IL-10) effect in treated mice [35].

#### 2.1.2. Bowel Inflammation

Oxidative stress increases damage to the mucosal layer in the gastrointestinal tract and stimulates an uncontrolled immune response, contributing to the initiation and development of inflammatory bowel disease (IBD) [73,74]. Ikumoto et al. found that MnSOD expression was elevated in the neutrophils, macrophages, and polymorphonuclear leukocytes in mucosal biopsy specimens of ulcerative colitis (UC) patients and characterized it as a potent diagnostic biomarker indicating a poor prognosis [75,76]. The engineered probiotic *Bifidobacterium longum* with a constructed expression system successfully secreted a biologically active penetratin-hMn-SOD fusion protein, which significantly inhibited ROS levels and cytokine release (TNF-α, IL-6, IL-1β, and IL-8), thus attenuating UC damage both in vitro and in vivo [77]. Consistently, *Lactobacillus gasseri* secreting MnSOD significantly inhibited inflammation in colonic tissues and alleviated IL-10-deficiency-induced colitis [78]. Moreover, rebamipide, an antiulcer agent, relieved mucosal injury and nonsteroidal anti-inflammatory drug (NSAID)-induced gastrointestinal complications such as gastric ulcers and erosions, by increasing MnSOD protein levels and suppressing superoxide anion leakage [45].

#### 2.1.3. Renal Inflammation

MnSOD has been demonstrated to play a protective role in maintaining kidney function. MnSOD-deficient mice exhibited increased oxidative stress, renal interstitial T cells, macrophage infiltration, tubular damage, and glomerular sclerosis compared with wild-type mice [79]. Additionally, albumin-overloaded mice exhibited decreased MnSOD protein levels and increased mitochondrial oxidative stress and inflammation in the kidney. Applying the MnSOD mimic Mn^III^ tetrakis (4-benzoic acid) porphyrin chloride (MnTBAP) rescued albumin overload-induced abnormalities by inhibiting angiotensin converting enzyme/angiotensin II signalling and renal sodium transporter expression [42,44].

#### 2.1.4. Rheumatic and Musculoskeletal Diseases

Rheumatic and musculoskeletal diseases (RMDs) can be divided into autoinflammatory diseases and autoimmune diseases [80]. Both complex inflammatory processes and activated innate and adaptive immunity contribute to the pathogenesis of RMDs [81,82]. 

Autoinflammatory diseases are a group of hereditary disorders that are characterized by seemingly unprovoked episodes of periodic fever and inflammation at certain body parts, and may even involve multiple organ systems [83]. Mevalonate kinase deficiency (MKD) is a rare paediatric disease caused by mutations in the mevalonate kinase (*MVK*) gene and the accumulation of mevalonolactone (MEV) metabolites [84]. Gratton et al. suggested that excess MEV disrupted mitochondrial functions and induced MnSOD gene expression in response to augmented ROS production in U-87 MG human glioblastoma cells [85]. Another inflammatory disorder of unknown aetiology, Behcet’s disease, is characterized by recurrent aphthous ulcers [86,87]. The Val/Val genotype of the MnSOD gene was significantly associated with Behcet’s disease in a group of Turkish patients [88]. These findings indicate that MnSOD is correlated with autoinflammatory diseases, but the detailed mechanisms remain largely unknown.

Autoimmune diseases, such as psoriasis and RA, involve errors in self-discrimination by adaptive immune mechanisms [89,90]. MnSOD was found to be overexpressed in the skin tissue of psoriasis patients, which protected against skin damage caused by elevated levels of the proinflammatory factors IL-1β and TNF-α [91,92]. RA is a chronic autoimmune disease characterized by the abnormal proliferation of fibroblast-like synoviocytes (FLSs), leading to synovial joint inflammation and joint deformities [93,94]. A comparative proteomics study revealed that MnSOD was significantly downregulated in FLSs from patients with RA [95]. Recombinant human MnSOD (rhMnSOD) administration ameliorated paw swelling and bone destruction in an adjuvant-induced arthritis rat model [60]. Bowen et al. constructed a nanosheet, Zn-Mn^III^ *meso*-tetrakis (4-carboxyphenyl) porphyrin-polyvinylpyrrolidone (ZMTP), to mimic MnSOD activity for RA treatment. They found that ZMTP mitigated oxidative stress in RA mice, decreased M1 macrophage markers (IL-6, IL-1β, and TNF-α), and promoted anti-inflammatory M2 phenotype markers (arginase-1 and IL-10) [41]. Seemingly contradictory, MnSOD was suppressed when RA-FLSs underwent mitochondrial-dependent apoptosis induced by resveratrol, indicating that the diverse roles of MnSOD in RA should be taken into consideration in future investigations [93]. 

### 2.2. Fibrotic Diseases

Fibrosis is defined by the excessive accumulation of extracellular matrix (ECM) components such as collagen and fibronectin in damaged tissues, which ultimately induce tissue sclerosis and organ failure [96]. This process is initiated by the continuous activation and transdifferentiation of local fibroblasts in response to chronic injury [97]. To date, the regulatory role of MnSOD in the fibrosis of multiple organs has been widely reported and summarized (Figure 1).

#### 2.2.1. Hepatic Fibrosis

Hepatic fibrosis is the result of a sustained wound healing response to continuous liver injury, which is initiated by the activation and transdifferentiation of hepatic stellate cells (HSCs) [98,99,100]. Platelet-derived growth factor (PDGF) is the most potent HSC mitogen and is responsible for intracellular ROS production. MnSOD upregulation impedes PDGF-induced proliferation and ROS generation in HSCs [101]. Certain natural compounds, such as curcumin [58] and alpha mangostin [59], exert promising inhibitory effects against the fibrogenesis process by increasing MnSOD expression and ROS scavenging to hinder HSC activation (Table 1). In addition, Guillaume et al. reported that rhMnSOD treatment significantly reduced hepatic oxidative stress and deactivated HSCs in CCl_4_-induced cirrhotic rats [61]. Moreover, hepatic fibrosis progression is commonly accompanied by non-alcoholic fatty liver disease (NAFLD) [102]. Several studies have examined the relationship between NAFLD and a functional polymorphism in the targeting sequence of MnSOD. In brief, carriage of the ‘T’ allele in the *SOD2* gene leads to less transport of MnSOD to the mitochondria, which is frequently found in non-alcoholic steatohepatitis (NASH) patients and is associated with the severity of liver fibrosis [103,104].

#### 2.2.2. Cardiac Fibrosis

Evidence has shown that MnSOD plays an essential role in cardiac fibrosis progression. MnSOD-deficient mice developed heart hypertrophy with enhanced fibrosis [105], while MnSOD overexpression inhibited ageing-induced cardiac fibrosis in transgenic mice [106]. These studies indicate that MnSOD expression negatively regulates the severity of cardiac fibrosis but the underlying mechanisms vary. For example, Ma et al. reported that the expression of mouse myocardial MnSOD was upregulated by AMPK after swimming exercises, thus attenuating isoproterenol-induced cardiac fibrosis via ROS clearance [107]. Vivar et al. reported that MnSOD mRNA and protein expression levels were reduced by transforming growth factor (TGF)-β1 in cardiac fibroblasts (CFs), resulting in elevated ROS levels in CFs [108]. Consistently, MnSOD-overexpressing transgenic aged mice exhibited a significant decrease in cardiac oxidative stress and fibrotic marker (TGF-β1, collagen I) expression in the heart [106]. These findings indicate a potential regulatory loop between MnSOD and TGF-β1 in the activation of CFs and subsequent cardiac fibrosis (Figure 2). Interestingly, melatonin increased MnSOD activity by promoting SIRT3-mediated MnSOD deacetylation to attenuate mitochondrial oxidative injury, ultimately alleviating fine particulate matter (PM_2.5_) exposure-induced cardiac dysfunction and fibrosis [46].

#### 2.2.3. Pulmonary Fibrosis

Pulmonary fibrosis is a common terminal syndrome in multiple lung diseases [109]. Since the lung is exposed to a hyperoxic environment and vulnerable to various oxidative pathogenic factors, MnSOD is especially vital for maintaining pulmonary redox balance to impede pulmonary fibrosis [110]. MnSOD activity was found to be significantly decreased in the fibrotic lung areas of patients with idiopathic pulmonary fibrosis (IPF). Further investigation showed that SIRT3 deficiency induced acetylation and the inactivation of MnSOD at lysine residue 68 exacerbated lung fibrosis in mice [111]. Moreover, a microarray indicated decreased MnSOD gene expression in systemic sclerosis interstitial lung disease (ILD) [112], and silencing MnSOD in fibroblasts derived from a non-ILD human resulted in a profibrotic phenotype [113]. Moreover, Zhang et al. reported that cigarette smoke extract induced lung fibrosis by downregulating MnSOD expression and enhancing collagen I expression in mice [114]. MnSOD is also recognized as a promising target for treating pulmonary fibrosis. Treatment with an rhMnSOD-Hirudin fusion protein effectively inhibited the proliferation of fibroblasts, suppressed profibrotic protein genes (*Ctgf*, *Fn*, *Col1a1*), and attenuated pulmonary fibrosis in vitro and in vivo [62].

#### 2.2.4. Fibrosis in Other Organs

A reduction in MnSOD expression was found in the kidneys of unilateral ureteric obstruction (UUO)-induced renal fibrosis mice [115,116], while Parkinson disease protein 7 (PARK7) protected against chronic kidney injury and fibrosis by stimulating MnSOD expression, thus reducing oxidative stress in both tubular cells and kidney tissues of UUO mice [117]. Rossi et al. discovered that TGF-β1/miRNA-382-5p/MnSOD axis deregulation is vital in the pathogenesis of myelofibrosis by enhancing oxidative stress and inflammation [118]. In addition, intra-arterial injection of lentiviral MnSOD (LVMnSOD) in rats protected skin tissues from fibrosis by reducing the fibrotic burden and skin contracture after irradiation [40].

### 2.3. Diabetes

Diabetes mellitus occurs whereby an abnormally high blood glucose level (hyperglycaemia) results from insulin insufficiency, and/or impaired tissue response to insulin (i.e., insulin resistance) [119]. It is a heterogeneous disorder with two major forms, type 1 diabetes mellitus (T1DM) and type 2 diabetes mellitus (T2DM), both of which are exacerbated by free radicals produced via glucose autoxidation [120,121,122]. Emerging evidence has indicated that the CT genotype of the MnSOD 47C/T gene is a significant risk factor for T1DM susceptibility (OR = 2.37; CI 95% = 1.03 to 5.46; *p* = 0.040) [123]. Here, we summarize the different roles of MnSOD in regulating diabetes and its complications.

#### 2.3.1. Type 1 and Type 2 Diabetes Mellitus

T1DM is an autoimmune disease triggered by the immune system attacking β-cells in the pancreatic islets, leading to β-cell dysfunction and hyperglycaemia [124]. T2DM, accounting for 85% of the diabetes burden, is caused by insulin resistance and inadequate compensatory insulin secretion, which is associated with several lifestyle factors, such as age, obesity, and pregnancy [125,126]. Smriti et al. found that MnSOD mRNA levels in the serum of newly diagnosed T2DM patients were obviously decreased (*p* = 0.02), indicating a correlation between MnSOD and T2DM [31]. In an STZ-induced T1DM mouse model, MnSOD-overexpressing pancreatic β-cells improved blood glucose by inhibiting NF-κB activation and scavenging ROS [127]. Sadi et al. demonstrated that treatment with vitamin C enhanced the mRNA level but not the enzyme activity of MnSOD, thus protecting diabetic tissues against oxidative stress [47]. Islet transplantation is a promising therapy for diabetes; however, the islet isolation and transplantation processes generate oxygen radicals and cause oxidative stress damage, which is a major obstacle to islet replacement therapy [128]. To prevent oxidative damage, Suzanne et al. isolated MnSOD-overexpressing islets from young prediabetic NOD mice and transplanted them into STZ-induced T1DM NOD *scid* mice. The survival rate of islet graft mice was increased, and MnSOD overexpression prolonged islet function after transplantation [129]. 

#### 2.3.2. Complications of Diabetes

Long-term hyperglycaemia causes diabetes complications that lead to damage throughout the body, which can be classified into microvascular and macrovascular damage [121,130,131]. Recent evidence has established that oxidative stress-regulated pathways in β-cell dysfunction and insulin resistance, collectively contribute to the development of diabetic complications, such as nephropathy, neuropathy, and retinopathy [132,133,134,135]. Oxidative stress has long been implicated in the progression of diabetes complications, leading to an increased risk of diabetic macrovascular and microvascular system lesions, such as diabetic cardiomyopathy and renal failure [133]. Increased MnSOD activity was observed as a self-defensive mechanism in the hearts of STZ-induced diabetic rats, indicating a protective role of MnSOD against oxidative stress-related damage in diabetic cardiomyopathy [136]. In addition, overexpression of MnSOD improved mitochondrial respiration and normalized contractility in diabetic cardiomyocytes by elevating cardiac catalase activity and preventing oxidative assaults [137]. Diabetic bladder dysfunction is another common diabetic consequence and has a profound impact on patient quality of life [138]. Excessive oxidative stress was observed in STZ-induced smooth muscle-specific MnSOD knockout mice, causing bladder dysfunction and apoptosis induction in detrusor smooth muscle, indicating a pivotal role of MnSOD in diabetic bladder dysfunction [139]. Moreover, MnSOD overexpression inhibited superoxide radicals and ameliorated the development of diabetic retinopathy by reducing glucose-induced mitochondrial DNA damage [140].

Taken together, these data highlight the involvement of MnSOD in regulating diabetes and its complications via oxidative stress inhibition.

### 2.4. Neurodegenerative Diseases

Neurodegenerative diseases are indicated by progressive degeneration and the selective loss of neuronal systems, leading to cognitive impairment, dementia, motor dysfunction, and even death [141]. The majority of neurodegenerative diseases are characterized by the accumulation of misfolded proteins in the central nerve system, e.g., amyloid-β (Aβ) and tau aggregates in Alzheimer disease (AD), SOD1 pathology in amyotrophic lateral sclerosis (ALS), and mutated huntingtin (HTT) in Huntington disease (HD), and accompanied by a progressive loss of neurons, e.g., the loss of dopaminergic (DA) neurons in PD (Parkinson’s disease) in the affected regions [142]. A number of studies have highlighted the potential therapeutic role of MnSOD in detoxifying cerebral ROS to inhibit the aggregation of misfolded proteins and protect against neurodegenerative disorders (Figure 1). 

#### 2.4.1. Alzheimer Disease

AD is a progressive neurodegenerative disorder characterized by impaired cognitive function and neuropsychiatric disorders. AD gradually destroys memory and thinking skills, and ultimately, disables the elderly from performing the simplest tasks [143,144,145]. Excessive accumulation of Aβ and phosphorylated tau are the main defining characteristics of AD [146,147]. Aβ peptides are generated from amyloid precursor protein (APP), which, together with presenilin 1 (PS-1) and PS-2, is suggested to be associated with the pathogenesis of early-onset AD [148]. It is well recognized that mitochondrial redox imbalance contributes to the neurodegeneration that occurs in AD; thus, MnSOD is involved in regulating AD progression, minimizing oxidative damage to the brain, and protecting cognitive function [149].

A decreased MnSOD level and an increased Aβ level were found in hippocampal neurons from autopsy-confirmed AD patients. This observation was also supported by spherical aggregation of Aβ in cultured human primary hippocampal neurons, which could be rescued by the MnSOD mimetic metalloporphyrin Mn^III^ tetrakis (1-methyl-4-pyridyl) porphyrin (MnTMPyP) [53,150]. Moreover, deletion of one copy of the MnSOD allele exaggerated cerebrovascular amyloidosis, amyloid burden, and plaque-independent neuritic dystrophy. Mechanistically, the reduction in MnSOD exacerbated endogenous mitochondrial oxidative stress and increased the level of Ser-396 phosphorylated tau in a mutant human APP transgenic AD mouse model [151,152,153]. Interestingly, overexpression of MnSOD significantly reduced plaque deposition and ameliorated learning and spatial memory deficits by scavenging mitochondrial ROS in a Tg2576/MnSOD double transgenic mouse model [154]. Another study suggested that 12 weeks of treadmill exercise could have beneficial effects on modulating Aβ deposition and cognitive function in the early stage of AD. The level of MnSOD enzyme activity in the hippocampus was significantly increased after exercise and attenuated microglia-mediated neuroinflammation in an APP/PS-1 mouse model [155].

However, MnSOD plays a paradoxical role in AD. Using an MnSOD-deficient *Saccharomyces cerevisiae* strain, Franca et al. verified that oxidative stress is a consequence of amyloid induction, suggesting that MnSOD may not be directly associated with the amyloid process [156]. Furthermore, Aβ-induced oxidative stress results in highly nitrated MnSOD in the peripheral mononuclear blood cells of AD patients and APP/PS-1 mouse models, which likely contributes to the neurodegeneration in AD pathogenesis [157,158].

In conclusion, MnSOD plays an important role in regulating oxidative stress and protecting the neural system and thus has attracted increasing attention as a therapeutic target and drug candidate for AD. 

#### 2.4.2. Parkinson’s Disease

PD is a multifactorial neurodegenerative disease characterized by the loss of DA neurons in the substantia nigra, resulting in a significant reduction in dopamine levels in the striatum [159]. Numerous studies have revealed a link between oxidative stress and DA cell function in PD [160]. As an antioxidant enzyme, MnSOD enzyme activity was reduced by dopamine-quinone (DAQ), which is generated during DA neuron degeneration and causes oxidative stress in PD. This reduction further exacerbated oxidative damage and subsequent neuronal dysfunction and eventually led to cell death [30]. Decreased MnSOD expression was also observed in PD-causing neurotoxin 1-methyl-4-phenyl-1,2,3,6-tetrahydropyridine (MPTP)-treated animal models, with severe loss of abnormal dopamine transporter (DAT)-immunoreactive DA neurons in the striatum and the substantia nigra [161,162]. This evidence indicates that MnSOD agonists may relieve PD by protecting DA neurons in the striatum and substantia nigra from oxidative stress. This has been confirmed by Dixit et al., who found that MnTMPyP could reverse MPTP-decreased MnSOD expression, thereby eliminating DA neuron loss in PD animals [54]. Moreover, increased MnSOD levels were found in the substantia nigra or frontal and motor cortex of PD patients, which could serve as an early diagnostic indicator [163,164,165]. Likewise, in two independent clinical trials, an increase in MnSOD mRNA levels was detected in 192 whole-blood samples from PD patients compared to healthy controls. A subsequent study suggested that the MnSOD mRNA level could be a potential blood biomarker for the diagnosis of PD patients [166]. A meta-analysis further revealed that the CC genotype of the MnSOD Val16Ala polymorphism indicated a higher susceptibility to PD in Han Chinese populations (OR = 1.77, 95% CI: 1.15–2.71, *p* = 0.01) [167]. 

#### 2.4.3. Huntington Disease

HD is an autosomal dominant, neurodegenerative disease manifesting as progressive chorea, rigidity, dementia, and psychiatric disturbance caused by accumulation of the mutant *HTT* gene in neural and somatic cells [168]. It has been widely reported that oxidative stress and mitochondrial dysfunction are closely linked to HD, suggesting MnSOD as a potent therapeutic target to ameliorate neurodegeneration [169,170,171]. Astrocytes, the most abundant cell type in the brain, are implicated in neurodegenerative disorders and may contribute to striatal neuron loss or dysfunction in HD. HD astrocytes with the accumulation of mutant *HTT* showed a marked decrease in the expression of MnSOD, which is consistent with previous reports on brain samples from HD patients [32,33]. Similarly, MnSOD activity was suppressed in the mitochondria of a 3-nitropropionic acid (3-NP)-induced HD rat model, while ROS production and lipid peroxidation were enhanced, accompanied by increased neural space, neurodegeneration, and gliosis. N-acetyl-L-cysteine (NAC) administration significantly attenuated neurobehavioural deficits and mitochondrial dysfunction by increasing MnSOD activity [48]. Phenotypic alterations in skin fibroblasts from HD patients are another feature associated with HD pathogenesis. For instance, significantly increased expression of the MnSOD gene was observed in primary porcine skin fibroblasts from a transgenic (TgHD) minipig HD model. Moreover, MnSOD enzyme activity was markedly elevated in skin fibroblasts of HD patients and may serve as a defender against antioxidant damage and a potential biomarker of HD [172,173,174].

#### 2.4.4. Amyotrophic Lateral Sclerosis

ALS is a rare and progressive neurodegenerative disease that is characterized by the degeneration of motor neurons in the brainstem and spinal cord, resulting in the loss of motor function and eventual death [175]. Although clinical evidence has confirmed that there is no significant correlation between MnSOD mutation and ALS, MnSOD can prevent SOD1 mutation-induced ALS [176]. For instance, MnSOD overexpression attenuated the neuronal cell death triggered by mutant SOD1^G37R^ in neuroblastoma cells [177]. Partial deficiency of MnSOD in ALS mice, induced by SOD1^G93A^ mutation, exacerbated motor neuron loss, motor deficits, and the death rate [178]. Increased MnSOD activity accompanied by excess superoxide production and lipid peroxidation was observed in the brain tissue homogenates of ALS rats [179]. Additionally, elevated MnSOD levels were observed in the motor neurons and glia of the brain stem in ALS patients, further supporting the proposed protective role of MnSOD in ALS [180,181].

### 2.5. Vascular Diseases

Redox homeostasis plays a vital role in vascular function. Dysregulated redox signalling can induce endothelial dysfunction and vascular abnormalities, contributing to different types of vascular diseases, such as hypertension and atherosclerosis (AS) and the occurrence of infarction [182,183,184]. Herein, we summarize the mechanisms of MnSOD-mediated protective effects through vascular remodelling in vascular diseases. 

#### 2.5.1. Hypertension

Hypertension is a disorder of circulatory regulation, with a persistent blood pressure elevation, and is listed as one of the major contributors to mortality [185]. In hypertension, abnormal oxidative stress increases both vascular stiffness and vascular smooth muscle cell adhesion, which further accelerates the development of endothelial dysfunction [186,187]. It has been reported that MnSOD in particular brain regions exerts antihypertensive effects by eradicating mitochondrial superoxide anions. For instance, MnSOD depletion in the brain subfornical organ (SFO) rather than in peripheral tissue significantly increased systemic mean arterial pressure and sensitized the pressor response, thus potentiating angiotensin II (AngII)-mediated hypertension in mice [188]. The gene expression and enzymatic activity of MnSOD were also found to be decreased in the brain stem rostral ventrolateral medulla (RVLM) of spontaneously hypertensive rats accompanied with elevated H_2_O_2_ and O_2_^•−^ level. Microinjections of adenovirus encoding MnSOD into the RVLM elicited a long-lasting reduction in arterial pressure [189]. In addition, the elevation of blood pressure and early marker of endothelial dysfunction (vascular cell adhesion molecule-1, VCAM-1) was markedly suppressed in MnSOD-transduced carotid and femoral arteries in a hypertensive rat model [190]. Moreover, dysfunction of endothelial progenitor cells leads to impaired endothelial integrity in hypertension patients. Treatment with mitoTEMPO, a MnSOD mimetic, reduced blood pressure and oxidative stress and improved the reendothelialization capability of endothelial progenitor cells in AngII-induced hypertension mice, providing a potential vascular repair therapeutic target for hypertension [55,56]. 

Recent studies have reported that mitochondrial oxidative stress induced by MnSOD deficiency contributes to the development of persistent pulmonary hypertension of the newborn (PPHN). Afolayan et al. found that silencing the MnSOD gene reproduced PPHN phenotypes, such as right ventricular systolic pressure elevation and pulmonary artery endothelial cell (PAEC) apoptosis [191]. Furthermore, exogenous transduction of PAECs with MnSOD markedly improved the function of endothelial nitric oxide synthase and the relaxation response of pulmonary arteries in PPHN lambs, supporting the use of MnSOD as a therapeutic target in PPHN treatments [191,192].

#### 2.5.2. Atherosclerosis

AS, a chronic disease defined as the formation of fibrofatty lesions in the artery wall, mainly results from pathologic processes characterized by vascular endothelial dysfunction and the formation of fibrous caps and plaques [193,194]. Emerging evidence has shown that MnSOD deficiency impairs vasomotor function and accelerates atherogenesis at arterial branch points [195,196]. Moreover, depletion of MnSOD increased the necrotic core, inflammatory cell infiltration, and the formation of atherosclerotic lesions in the arteries of an apolipoprotein E-deficient (ApoE^-/-^) spontaneous AS mouse model [57,197]. In addition, chronic inflammatory skin diseases, such as psoriasis, have been reported to accelerate AS. A significant reduction in MnSOD expression was found in psoriatic macrophages, while restoring MnSOD activity with MnTBAP reversed the proatherosclerotic phenotype in a mouse model [52]. Accordingly, elevating MnSOD activity could be a therapeutic approach to retard AS progression. For instance, treatment with the MnSOD mimetic mitoTEMPO attenuated atherosclerotic plaque vulnerability by preventing intimal smooth muscle cell apoptosis and matrix degradation in a mouse model [57]. The mitochondrial protective agent idebenone significantly stabilized atherosclerotic plaques in ApoE^-/-^ mice by activating the SIRT3-MnSOD-mtROS pathway [49]. Treatment with hawthorn fruit extract significantly upregulates MnSOD expression, thus reducing trimethylamine-N-oxide-exacerbated atherogenesis in ApoE^−/−^ mice [49]. 

#### 2.5.3. Myocardial/Cerebral Infarction and Reperfusion Injury

Myocardial/cerebral infarction usually occurs when an atherosclerotic plaque hinders the blood supply in the heart or brain [194]. As a common therapy in the clinic, revascularization causes secondary damage to ischaemic tissues when blood flow is restored, which is called ischaemia/reperfusion (I/R) injury. MnSOD expression was elevated by an endogenous cardioprotective enzyme, dimethylarginine dimethylaminohydrolase 1 (DDAH1), to attenuate acute myocardial infarction (MI) by suppressing apoptosis and oxidative stress in cardiomyocytes [198]. Asprosin, a novel MI therapeutic agent, inhibited mesenchymal stromal cell apoptosis by activating extracellular signal-regulated kinase 1/2 (ERK1/2)-MnSOD signalling [50]. Furthermore, MnSOD is a key enzyme that plays a protective role after myocardial I/R injury. Various therapeutic agents used to treat I/R injury (Table 1), such as sevoflurane, trans sodium crocetinate, and irisin, were found to elevate MnSOD levels to protect cardiomyocytes from oxidative stress and apoptosis [36,37,38]. Moreover, MnSOD overexpression was verified to reduce the infarct heart size in a left coronary artery ligation I/R model in MnSOD transgenic mice [199].

Cerebral infarction (ischaemic stroke), defined as the sudden loss of focal neurological function due to cerebral ischaemia, is pathologically induced by hypertension, carotid stenosis, AS, or atrial fibrillation [200]. Zhao et al. reported that MnSOD was significantly elevated by miRNA-518-5p/miRNA-3135b signalling based on single-cell and bulk RNA-sequencing analyses, and this result was further validated in the ischaemic cerebral infarction group in a clinical cohort study [201]. An in vitro study demonstrated that a contraceptive drug, Nestorone, could ameliorate behavioral and histological stroke outcomes by activating MnSOD in neural stem cells [51]. Pretreatment with MnTMPyP significantly eliminated intracellular superoxide radical levels, and an in vivo study showed that it reduced cerebral infarct size and improved neurological function [43]. Moreover, elevated MnSOD expression in the cortex was demonstrated to ameliorate cerebral I/R-induced oxidative stress injury via several miRNAs (miRNA-424, miRNA-23a-3p, and miRNA-182) in a middle cerebral artery occlusion (MCAO) and reperfusion mouse model [202,203,204]. In addition, electroacupuncture pretreatment upregulated MnSOD expression to inhibit cellular apoptosis in the ischaemic penumbra in MCAO mice. Interestingly, the beneficial effects of electroacupuncture were reversed by MnSOD knockdown, indicating the unique role of MnSOD in protecting against I/R injury [205].

### 2.6. Cancer

Various aspects of the extensive role of MnSOD in cancer have been explored. MnSOD can directly mediate multiple cancer cell death signalling pathways, including apoptosis [206], pyroptosis [207], and autophagy [208]. High MnSOD expression contributes to chemoresistance [209,210,211] and radioresistance [212,213] in different cancer types. Recently, posttranslational modification of MnSOD, with particular emphasis on acetylation at lysine residue 68, was proposed and explored, which addressed its vital roles in promoting cancer progression [214,215,216]. These functional roles of MnSOD in cancer have been well-reviewed elsewhere [12,27,217,218]. Herein, we summarize some emerging research hotspots in MnSOD-regulated cancer progression, with an emphasis on metabolic reprogramming and the tumour immune microenvironment.

Metabolic reprogramming has been regarded as a hallmark of cancer for centuries. The Warburg effect primarily demonstrates that malignant cells preferentially rely on glycolysis rather than oxidative phosphorylation (OXPHOS) for energy supply [27,219,220]. Moreover, pathways involved in redox control are commonly reprogrammed due to tumorigenic mutations [221]. As MnSOD is one of the major redox metabolism regulators, the effect of MnSOD on cancer metabolic reprogramming has attracted broad attention. For instance, Liu et al. reported that toll-like receptor 2 (TLR 2) induced glycolysis and metabolic shift by markedly upregulating MnSOD expression in human gastric cancer cells. Furthermore, patient-derived tissue microarrays demonstrated a significant correlation between TLR2 and MnSOD expression in gastric tumours, indicating that the TLR2-MnSOD axis could serve as a potential biomarker for therapy decisions and prognosis determination in gastric cancer [222]. Zhou et al. observed that enhanced MnSOD expression activated AMPK signalling and shifted cell energy metabolism to glycolysis, thus facilitating tumorigenesis and progression in colorectal cancer cells [223]. Similarly, upregulation of MnSOD in cancer cells engaged AMPK to perform and sustain the Warburg effect, therefore supporting cancer cell survival [224]. Cells with a high level of MnSOD were more resistant to glucose-deprivation-induced cell death because MnSOD increased glucose uptake, glucose transporter member 1 (GLUT-1) availability, and OXPHOS electron transfer [225]. These results indicate a vital role of MnSOD in cancer glycolytic metabolism reprogramming, which deserves more in-depth mechanistic studies. 

As the role of the immune system in cancer development has attracted increasing attention recently, the involvement of MnSOD in the tumour immune microenvironment has also been highlighted. Accumulating evidence has suggested that tumour-infiltrating immune cells participate in cancer progression, which is highly associated with MnSOD expression in different cancer types. For example, Su et al. analysed public datasets and found that MnSOD expression is positively correlated with CXCL8 and neutrophil infiltration, indicating an involvement of the ‘MnSOD-CXCL8-neutrophil recruitment’ axis in cancer progression [226]. MnSOD was also found to be positively correlated with CD68+ macrophage infiltration and may indicate poor outcomes in inflammation-driven lung adenocarcinoma [227]. High MnSOD expression was observed in aggressive triple-negative breast cancer (TNBC) patients and was regarded as a poor prognostic marker. MnSOD promotes the immunosuppressive tumour microenvironment by promoting M2 macrophage invasiveness and infiltration, supporting TNBC progression [228]. In addition, Lou et al. demonstrated that MnSOD-overexpressing oncolytic vaccinia virus (OVV-MnSOD) enhanced lymphocyte infiltration and tumour sensitivity to anti-PD-L1 treatment in lymphomatous mice, suggesting a promising therapeutic potential of MnSOD in cancer immunotherapy [63].

In addition, we recently explored whether MnSOD overexpression could prevent cancer relapse by inhibiting reactivation of quiescent cancer cells (data not published). This novel function of MnSOD further increases the potential benefits of targeting MnSOD in cancer treatment. Although the pleiotropic role of MnSOD in cancer progression has aroused extensive interest, the underlying mechanisms remain controversial and unclear. Breakthroughs in finding key players in MnSOD-cancer regulation will be a promising research area, and will provide positive insights into the development of new drugs based on MnSOD.

## 3. Discussion and Future Prospects

In this review, we described preclinical and clinical evidence on the involvement of MnSOD in inflammatory, fibrotic, neurodegenerative, and vascular diseases, as well as diabetes and cancers. Mechanistically, MnSOD mainly protects the host from oxidative stress by maintaining the redox homeostasis when exposed to stressors such as radiation, ultraviolet, chemotherapy, hypoxia, and hyperglycaemia [9,229,230,231]. Therefore, targeting MnSOD provides a promising strategy for preventing and treating redox-imbalance-related diseases.

However, several limitations and critical issues in these MnSOD-related studies remain to be elucidated. Though the above studies revealed that MnSOD is abnormally expressed in different diseases, whether these aberrant expressions are the cause or consequence of diseases remains undiscovered. Such a cause–effect relationship between MnSOD and diseases may largely determine the appropriateness of an MnSOD-targeting therapeutic approach. This point extends to another crucial question, whether oxidative stress is the principal pathogenesis in the above diseases? However, to date, the effectiveness of antioxidant defence is limited because oxidative stress is more often the secondary cause, rather than becoming the primary cause of disease [13]. Such an uncertain situation challenges the MnSOD-targeted strategy in effectively ameliorating disease. Exploring which oxygen-derived free radicals and/or molecule that principally trigger pathological consequences deserves serious consideration in stepping forward in this field. To this regard, applying the recent multi-omics and imaging technologies may provide comprehensive understanding on the above issue, to facilitate the identifying of impaired steps in redox metabolism, and ultimately, to evaluate the therapeutic effects of MnSOD-targeting agents.

Although few limitations in understanding the pathological role of MnSOD remain, emerging evidence has supported its broad clinical applicability and potential as a prognostic biomarker. For example, melatonin treatment (NCT02463318) attenuated oxidative stress by directly increasing the gene expression and enzyme activity of MnSOD in peripheral blood mononuclear cells (PBMCs) of relapsing–remitting multiple sclerosis (RRMS) patients [232]. A phase I-II study (NCT00618917) demonstrated that a swallowed MnSOD-plasmid/liposome (PL) transgene treatment alleviated radiation/chemotherapy-induced oesophagitis in advanced stage III non-small cell lung cancer patients. Moreover, it was found that MnSOD activity in the PBMCs of patients was linked to reduced diabetes- and cardiovascular-related complications [233,234]. After overall high-intensity training, T1DM patients (NCT02939768) showed increased MnSOD activity in mononuclear cells. A case–control study found that the MnSOD level in the PBMCs of patients could well reflect the therapeutic effect of resveratrol against coronary artery disease (NCT02137421). 

MnSOD mimetics have recently gained increased attention, and a series of synthetic molecules were found to exhibit better uptake and ROS clearance efficiency [235,236]. For instance, M40403, a manganese poly-aza-macrocycle, synergistically enhanced the analgesic effects of morphine in postoperative dental pain [235]. Unfortunately, clinical trials investigating M40403 for cancer treatment have either been suspended (NCT00033956) or terminated (NCT00101621). MnBuOE, a type of manganese porphyrin, was used as a radioprotector of brain tissues in glioma patients (NCT02655601). In addition, MnBuOE was also studied in another phase II clinical trial to examine its protective effect against radiation-induced mucositis and xerostomia (NCT04607642). Furthermore, M40403 (NCT00033956), Tempol (NCT01324141), and calmangafodipir (NCT01619423) were studied in clinical trials as adjuvant agents to treat melanoma, renal cell carcinoma, anal cancer and advanced metastatic colorectal cancer. A preclinical study demonstrated that GC4419, a cyclic polyamine type of MnSOD mimetic, ameliorated the inflammatory response in chronic obstructive pulmonary disease [237]. GC4419 was studied in a phase II trial in patients with critical illness due to COVID-19 infection (NCT04555096) and might be a promising treatment for patients at risk of developing life-threatening SARS-CoV-2 [238].

Moreover, recent emphasis has been placed on reforming MnSOD or its mimetics to increase the efficiency and accuracy of tissue-specific delivery. An engineered recombinant MnSOD (rMnSOD) protein containing a leader peptide (rMnSOD-Lp) exhibited better cancer cell penetration ability [229,239]. This advantage also helps rMnSOD-Lp serve as a molecular carrier that can be conjugated with chemotherapeutic agents, such as cisplatin, and selectively delivered to cancer cells [240,241]. On the other hand, coupling MnSOD or its mimetics with nanoassemblies has shown success in potentiating therapeutic efficacies. A constructed two-dimensional nanosheet catalyst with an MnSOD mimetic (manganese porphyrin MnTCPP^+^), ZMTP, provided a much better and longer-lasting anti-inflammatory effect in RA than MnTCPP^+^ treatment alone. Further mechanistic studies revealed that ZMTP nanosheets significantly elevated catalytic activities accompanied by electron and proton transfers [41]. The incorporation of rMnSOD with a polymer multilayer nanocapsule significantly delayed rMnSOD denaturation, enhanced cytotoxicity, increased recognition of cluster of differentiation 44 (CD44), and prolonged the residence time in cancerous lesions [242]. Recently, facilitating MnSOD mimetics to enter mitochondria is a promising approach for better therapeutic efficiency. By conjugating MnSOD mimetics with lipophilic cations, the large negative inner mitochondrial membrane potential navigates these antioxidant mimics to mitochondria [238]. Nevertheless, these mitochondria-specific agents might be cytotoxic as they interfere the production of adenosine triphosphate (ATP), which should be taken into consideration in future investigations.

In conclusion, the expanding biological functions and therapeutic potentials of MnSOD have already been highlighted in numerous diseases. Broad discussion of the multiple aspects of MnSOD will shed light on the unknown mechanisms that govern redox-related diseases and provide optimized strategies for therapeutic intervention.

## Figures and Tables

**Figure 1 ijms-23-15893-f001:**
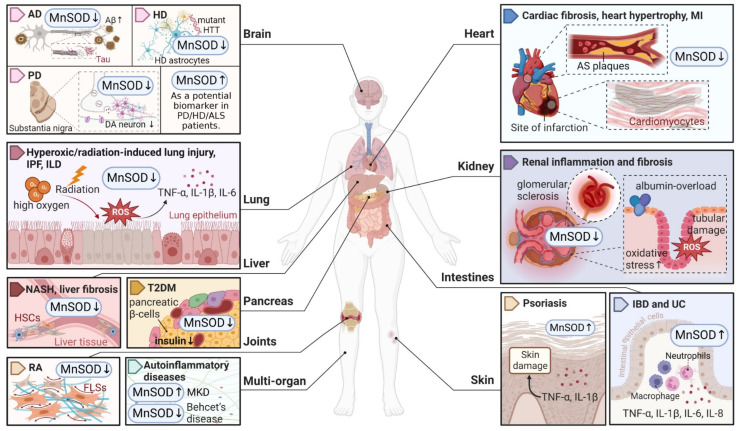
An overview of MnSOD expression in various pathological states of human diseases. Abnormal expression levels of MnSOD have been detected in multiple diseases and have a wide impact on different organs, followed by various pathological alterations such as inflammation, fibrosis, sclerosis, and neuron degeneration. MnSOD, manganese superoxide dismutase; ROS, reactive oxygen species; AD, Alzheimer’s disease; Aβ, amyloid β-protein; HD, Huntington’s disease; HTT, huntingtin; PD, Parkinson’s disease; DA, dopaminergic; ALS, amyotrophic lateral sclerosis; IPF, idiopathic pulmonary fibrosis; ILD, interstitial lung disease; TNF, tumour necrosis factor; IL, interleukin; NASH, non-alcoholic steatohepatitis; HSCs, hepatic stellate cells; T2DM, type 2 diabetes mellitus; RA, rheumatoid arthritis; FLSs, fibroblast-like synoviocytes; MKD, mevalonate kinase deficiency; MI, myocardial infarction; AS, atherosclerosis; IBD, inflammatory bowel disease; UC, ulcerative colitis. ‘↑’ represents increased or upregulated, while ‘↓’ represents decreased or downregulated (created with BioRender.com (accessed on 4 November 2022)).

**Figure 2 ijms-23-15893-f002:**
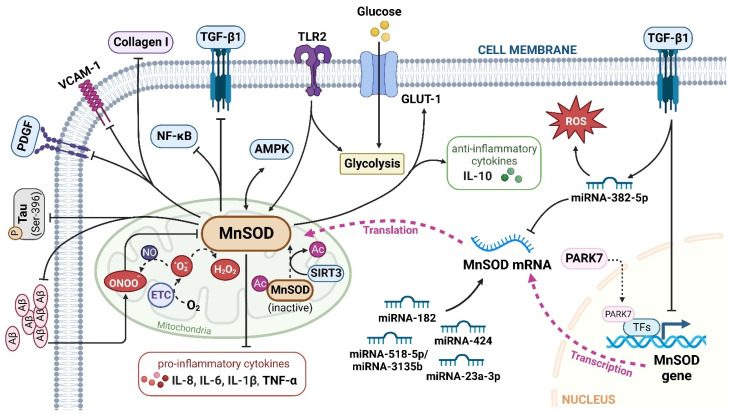
The major signalling pathways involved in MnSOD regulation. MnSOD is localized in the mitochondrial matrix to catalyse the dismutation of O2^•−^ to H2O2, and regulate cellular redox homeostasis. Multiple factors, cytokines, proteins, and miRNAs have been involved in the transcriptional and translational modulation of MnSOD. TGF-β1, transforming growth factor-β1; TLR2, toll-like receptor 2; Aβ, amyloid, β-protein; GLUT-1, glucose transporter member 1; IL, interleukin; VCAM-1, vascular cell adhesion molecule-1; AMPK, adenosine 5′-monophosphate-activated protein kinase; P, phospho; PDGF, platelet-derived growth factor; NF-κB, nuclear factor kappa-B; ETC, electron transport chain; PARK7, Parkinson disease protein 7; TFs, transcriptional factors; Ac, acetyl; SIRT3, sirtuin 3. ‘↑’ represents increased or upregulated, while ‘⊥’ represents suppressed or downregulated (created with BioRender.com (accessed on 4 November 2022)).

**Table 1 ijms-23-15893-t001:** MnSOD-based therapeutic strategies.

Type	Treatment	Chemical Structures	Disease	Mechanism	Therapeutic Effect	Ref
Clinical agents	Rebamipide	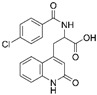	Bowel inflammation	↑MnSOD protein levels↓superoxide anion leakage	↓NSAID-induced mucosal injury	[45]
Melatonin	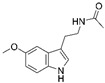	Cardiac fibrosis	↑MnSOD activity↑SIRT3-mediated MnSOD deacetylation	↓cardiac dysfunction and fibrosis	[46]
Vitamin C	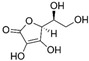	Diabetes	↑MnSOD mRNA levels	protected diabetic tissues	[47]
NAC	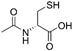	HD	↑MnSOD activity↓mitochondrial dysfunction	↓neurobehavioural deficits	[48]
Idebenone	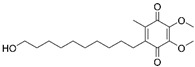	AS	↑SIRT3-MnSOD-mtROS pathway	stabilized atherosclerotic plaques	[49]
Asprosin	N/A	Ischaemic heart disease	↑ERK1/2-MnSOD signalling↓mesenchymal stromal cell apoptosis	↑function of cardiac ejection↓MI-induced myocardial remodelling	[50]
Nestorone	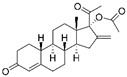	Ischaemic stroke	↑MnSOD protein levels	↓behavioural and histological stroke outcomes	[51]
Sevoflurane	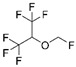	Myocardial I/R injury	↑MnSOD activity and protein levels	↓myocardial I/R injury	[36,37,38]
Trans sodium crocetinate	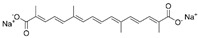
Irisin	N/A
MnSOD mimetics	ZMTP	N/A	RA	↑transition of M1 macrophages to M2 phenotype↓IL-6, IL-1β and TNF-α↑arginase-1 and IL-10	↑antiarthritic efficacy	[41]
MnTBAP	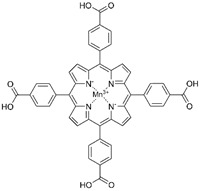	Kidney disease	↓angiotensin converting enzyme/angiotensin II signalling ↓mitochondria-derived oxidative stress	↓albumin-overload-induced abnormalities	[42,44]
AS	reversed the proatherosclerotic macrophage phenotype	↓AS progression	[52]
MnTMPyP	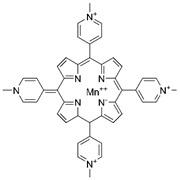	AD	↓ROS during neurodegeneration	protected hippocampal neurons	[53]
PD	↑MnSOD protein levels↓DA neurons loss	↓nigrostriatal DA neurodegeneration	[54]
Ischaemic stroke	↓intracellular superoxide radical levels	↓cerebral infarct volume↑neurological function	[43]
mitoTEMPO	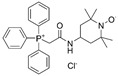	Hypertension	↑reendothelialization capability of endothelial progenitor cells	↓blood pressure↑vasorelaxation	[55,56]
AS	↓intimal smooth muscle cell apoptosis and matrix degradation	↓features of atherosclerotic plaque vulnerability	[57]
Natural compounds	Curcumin	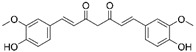	Hepatic fibrosis	↑MnSOD activity and mRNA levels↓HSC activation and ROS	↓fibrogenesis	[58,59]
Alpha mangostin	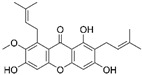
Hawthorn fruit extract	N/A	AS	↑MnSOD mRNA levels	↓atherogenesis	[49]
Proteins	rhMnSOD	N/A	RA	↑plasma concentration of MnSOD	↓paw swelling and bone destruction	[60]
Hepatic fibrosis	↓HSC activation	↓hepatic fibrosis	[61]
rhMnSOD-Hirudin fusion protein	N/A	IPF	↓fibroblasts proliferation↓profibrotic genes (*Ctgf*, *Fn*, *Col1a1*)	↓lung inflammation and fibrosis	[62]
Oncolytic vaccinia virus	OVV-MnSOD	N/A	Lymphoma	↑lymphocyte infiltration↑tumour sensitivity to anti-PD-L1 treatment	↓tumour progression	[63]

Note: mtROS, mitochondrial reactive oxygen species; NSAID, nonsteroidal anti-inflammatory drug; SIRT3, sirtuin 3; NAC, N-acetyl-L-cysteine; HD, Huntington’s disease; PD, Parkinson disease; AD, Alzheimer’s disease; PD-L1, programmed death ligand 1; ERK1/2, extracellular signal-regulated kinase 1/2; MI, myocardial infarction; I/R, ischaemia/reperfusion; IL, interleukin; TNF, tumour necrosis factor; DA, dopaminergic; HSC, hepatic stellate cell; OVV, oncolytic vaccinia virus; AS, atherosclerosis; RA, rheumatoid arthritis; IPF, idiopathic pulmonary fibrosis; ZMTP, Zn-Mn^III^ *meso*-tetrakis (4-carboxyphenyl) porphyrin-polyvinylpyrrolidone; MnTBAP, Mn^III^ tetrakis (4-benzoic acid) porphyrin chloride; MnTMPyP, Mn^III^ tetrakis (1-methyl-4-pyridyl) porphyrin. ‘↑’ represents increased, upregulated, induced, enhanced, or activated, while ‘↓’ represents decreased, downregulated, or inhibited, and ‘N/A’ represents not applicable.

## Data Availability

Not applicable.

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
