# Peer review of "Insights into Manganese Superoxide Dismutase and Human Diseases"

_ijms, 2022, doi:10.3390/ijms232415893_

Round 1

Reviewer 1 Report

In this manuscript, the authors indicate Insights into MnSOD and human diseases. The manuscript can be accepted after addressing the below mentioned corrections.

1.     It would be better if the long name of MnSOD was written in the title.

2.     In introduction, After sentences, ‘The dynamic balance between reactive oxygen species (ROS) production and antioxidant decomposition maintains cellular redox homeostasis..’’ It should be given information. For this purpose the authors can look at the following articles for introduction section: Journal of pharmacy and pharmacology 71 (10), 1576-1583 and Drug development research 81 (5), 628-636.

3.     In introduction, After sentences, ‘’As the major sites of ROS production, mitochondria are a principal source of cellular superoxide generation, which 31 is prone to induce oxidative damage and stress.’’ It should be given information. For this purpose the authors can look at the following articles for introduction section: Pharmacological reports 71 (3), 545-549 and Environmental toxicology and pharmacology 70, 103195.

4.     After sentences, ‘Excess ROS can induce mitochondrial dysfunction, inflammation and apoptosis, leading to the pathogenesis of multiple disor-33 ders and organismal damage.’’ It should be given information. For this purpose the authors can look at the following articles for introduction section: Comparative biochemistry and physiology part C: toxicology & pharmacology 226, 108608 and Applied biochemistry and biotechnology 190 (2), 437-447.

5.     In 2.3 section, After sentences, ‘Diabetes mellitus is characterized by high blood glucose levels (hyperglycaemia) resulting from insulin insufficiency’’ It should be given information. For this purpose the authors can look at the following articles for introduction section:Open Chemistry 19 (1), 347-357 and Drug Development Research 83 (3), 586-604.

6.     In 2.3.2. section, After sentences, Diabetes and its complications are complex, multifactorial diseases that cause long term damage throughout the body. It should be given information. For this purpose the authors can look at the following articles for introduction section: Archives of physiology and biochemistry 128 (4), 979-984 and Journal of Molecular Recognition 35(12), e2991

7.     In 2.3.2. section, After sentences, Oxidative stress has long been implicated in the progression of diabetes complications, leading to an increased risk of diabetic macro vascular and microvascular system lesions, such as diabetic cardiomyopathy and renal failure. It should be given information. For this purpose the authors can look at the following articles for introduction section: Journal of Molecular Structure 1258, 132675 and Chemico-Biological Interactions 345, 109576.

8.     In 2.4. section After sentences, Neurodegenerative diseases are characterized by progressive degeneration and selective loss of neuronal systems, leading to cognitive impairment, dementia, motor dysfunction, and even death. It should be given information. For this purpose the authors can look at the following articles for introduction section: Archiv der Pharmazie 354 (12), 2100294 and Molecular Diversity, 26, 2825–2845

9.     In 2.4.1 section After sentences, AD is a progressive neurodegenerative disorder characterized by impaired cognitive function and neuropsychiatric disorders. It should be given information. For this purpose the authors can look at the following articles for introduction section: Arabian Journal of Chemistry 15 (3), 103645 and Archives of physiology and biochemistry 125 (3), 235-243.

10.  There are several English language issues comprising poorly structured sentences and syntactic errors.

Author Response

  1. It would be better if the long name of MnSOD was written in the title.
    Response: Agreed. We have changed ‘MnSOD’ to ‘manganese superoxide dismutase’.
  2. In introduction, After sentences, ‘The dynamic balance between reactive oxygen species (ROS) production and antioxidant decomposition maintains cellular redox homeostasis.’’ It should be given information. For this purpose the authors can look at the following articles for introduction section: Journal of pharmacy and pharmacology 71 (10), 1576-1583 and Drug development research 81 (5), 628-636. In introduction, After sentences, ‘As the major sites of ROS production, mitochondria are a principal source of cellular superoxide generation, which is prone to induce oxidative damage and stress.’ It should be given information. For this purpose the authors can look at the following articles for introduction section: Pharmacological reports 71 (3), 545-549 and Environmental toxicology and pharmacology 70, 103195. After sentences, ‘Excess ROS can induce mitochondrial dysfunction, inflammation and apoptosis, leading to the pathogenesis of multiple disorders and organismal damage.’’ It should be given information. For this purpose the authors can look at the following articles for introduction section: Comparative biochemistry and physiology part C: toxicology & pharmacology 226, 108608 and Applied biochemistry and biotechnology 190 (2), 437-447.
    Response: Thanks for the suggestion. We have rearranged the introduction section (as requested by Reviewer 2 and added the relevant information and references.
  3. In 2.3 section, After sentences, ‘Diabetes mellitus is characterized by high blood glucose levels (hyperglycaemia) resulting from insulin insufficiency’’ It should be given information. For this purpose the authors can look at the following articles for introduction section:Open Chemistry 19 (1), 347-357 and Drug Development Research 83 (3), 586-604. In 2.3.2. section, After sentences, Diabetes and its complications are complex, multifactorial diseases that cause long term damage throughout the body. It should be given information. For this purpose the authors can look at the following articles for introduction section: Archives of physiology and biochemistry 128 (4), 979-984 and Journal of Molecular Recognition 35(12), e2991. In 2.3.2. section, after sentences, Oxidative stress has long been implicated in the progression of diabetes complications, leading to an increased risk of diabetic macrovascular and microvascular system lesions, such as diabetic cardiomyopathy and renal failure. It should be given information. For this purpose the authors can look at the following articles for introduction section: Journal of Molecular Structure 1258, 132675 and Chemico-Biological Interactions 345, 109576.
    Response: We have added relevant references in the 2.3 section according to the reviewer’s suggestion in the revised manuscript.
  4. In 2.4. section After sentences, Neurodegenerative diseases are characterized by progressive degeneration and selective loss of neuronal systems, leading to cognitive impairment, dementia, motor dysfunction, and even death. It should be given information. For this purpose the authors can look at the following articles for introduction section: Archiv der Pharmazie 354 (12), 2100294 and Molecular Diversity, 26, 2825–2845. In 2.4.1 section After sentences, AD is a progressive neurodegenerative disorder characterized by impaired cognitive function and neuropsychiatric disorders. It should be given information. For this purpose the authors can look at the following articles for introduction section: Arabian Journal of Chemistry 15 (3), 103645 and Archives of physiology and biochemistry 125 (3), 235-243.
    Response: We have added relevant references in the 2.4 section as suggested.
  5. There are several English language issues comprising poorly structured sentences and syntactic errors.
    Response: We have modified the English language in the revised manuscript.

Reviewer 2 Report

Mengfan Liu et al present here a review on MnSOD and its role in several diseases. The authors also describe the effect of certain mimics of MnSOD or therapeutics in these same pathologies. The review begins with a very brief introduction accompanied by a figure on the state of expression of MnSOD in certain pathologies and a nice descriptive table of the various existing pharmaceutical strategies, their mechanism and therapeutic effects. All this is followed by a systematic LISTING of the intervention of MnSOD or its mimics or drugs in several pathologies and is completed by a new figure representing the different pathways regulated by MnSOD (or more accurately by the levels of superoxide radical anions, H2O2 or peroxynitrite).

This referee has raised several points that need to be resolved before this review is even potentially considered for publication. Most of them are listed below:   MAJOR POINTS:   - The most important point: this referee thinks that making a more or less descriptive catalog of all the pathologies in which MnSOD appears is not read willingly and sometimes does not add anything more to what is already present in the literature. Thus, a restructuring of this review with a new introduction (ROS, ROS signaling, MnSOD – see below), a summary of the studies carried out (more or less described in the table) followed by an answer on emerging research questions on the analyzed topic and a conclusion of what can be developed and where we can hope to go next (roughly written is the last paragraph) would be preferable to a descriptive catalog. As presented, this referee feels that this review does not provide a clear and comprehensive overview of scientific advances in this field.   - The article would have clearly gained in clarity and impact if this review contained a clear and concise introductory paragraph on Reactive Oxygen Species (ROS), their definition, their genesis, their signaling functions under normal conditions and the involvement (among others) of MnSOD in these signaling functions. Indeed, it is now well-known that mitochondria drive +/- cellular functions by modulating the production of metabolites intervening as signaling molecules. mtH2O2 is essential for energetic/metabolic homeostasis and H2O2 drives cellular metabolism under physiological conditions. Thus, the question of how H2O2 is produced and regulated in the mitochondria is crucial. And H2O2 formation in mitochondria depends on and is controlled by MnSOD. The reader should at least know these points. There are many excellent articles that demonstrate these findings.   - In line with the previous comment, the authors state that “MnSOD is a principal antioxidant enzyme that scavenges excessive ROS” in the abstract. This is a serious error. ROS include but are not exclusively superoxide radical anions! Furthermore, MnSOD converts O2°- into H2O2 AND O2 (do not forget O2, please).   - On the other hand, a more thorough paragraph on MnSOD must be presented in this review. It could deal with its biogenesis, its metallation process then assembly into its quaternary structure (see for instance articles from Valeria Culotta) and its regulation (acetylation at K68, phosphorylation at Ser106, regulation via the mitochondrial protein influx...) as well as the place and the effects of this regulation on the properties catalytic activity, stability...) and functions of the enzyme. In addition, when describing metallation, a brief allusion to the replacement of Mn2+ by Fe2+ in cases of Mn2+ deficiency could be made, this leading to a pro-oxidant function of the enzyme via a peroxidase activity.   - The above paragraph could be briefly completed by the various signals that leads to MnSOD down-regulation or up-regulation (instead of its overexpression thanks to lenlentivirus or transgene) as well as by the various modifications/inhibitions of MnSOD observed in (pathological) conditions such as nitration by peroxynitrite, product-inhibited complex…  This would make it possible to establish a better connection with all the pathologies addressed by the authors in which MnSOD is involved.   - According to this referee, the authors make some small “shortcuts”, approximations, in the introductory paragraph and the abstract. For example, the authors write that "these observations indicate that MnSOD is widely involved in multiple biological processes...". Nevertheless, is it really MnSOD or variations in the concentrations of its substrate (O2°-) or products (in particular H2O2 and variation of its signaling)? Also, “The SOD family is in the front line of antioxidant enzymes that defend against ROS”. And yet, SOD produces ROS (H2O2) whose high concentrations are pathogenic... Hence, the necessity of a more thorough introductory paragraph on ROS, MnSOD and the other enzymes that are the first bastion of defense against ROS (catalase (CAT) and glutathione peroxidase (GPx)). In addition, what are “ROS-related diseases” (abstract)? Are ROS the source of the diseases, the consequence…? It reads as if ROS were responsible for the listed diseases whereas ROS are mediators of cellular damage in the described diseases including for example diabetes in which acute or chronic high glucose increases the production of ROS.   - Finally, the article contains numerous errors regarding references that are not properly adequate or are misused. For instance, in the introductory paragraph, ref. 1 deals specifically with mtROS and mtROS signaling and is thus inappropriate to describe the dynamic balance between ROS production/degradation and the maintenance of cellular redox homeostasis. Ref. 10 discusses the use of MnSOD and SOD mimics as radioprotectors and is therefore unsuitable to describe its strategic location and its role in superoxide dismutation during cellular respiration and its function in the maintenance of redox homeostasis. Ref. 11 deals with increased acetylated levels of MnSODK68 in patients harboring IPF. Accordingly, MnSOD gene expression is not downregulated in this case. Similarly, ref. 13 debates about MnSOD as a target of DAQ with a resulting 50% loss in its enzymatic activity. No downregulation in here too. Unfortunately, this point (references that are not truly adequate or misplaced) is recurrent throughout the article… For example, paragraph 2.1.1 begins with ref. 54 that only and solely discusses the treatment of a rat model of lung pleurisy with the MnSOD mimetics MnTBAP! Hence, only one model of lung damage and not “different modelS” as stated by the authors. However, refs. 17,18,54 and 55 (altogether) showed indeed that “MnSOD was found to attenuate pulmonary inflammation in different models of lung damage”. Then a very serious error is found in paragraph 2.1.4 when the authors state that “the elevation of MnSOD activity was also observed in plasma” (ref. 69). But the major SOD isoenzyme in plasma is extracellular SOD (EC-SOD) that is mostly secreted by endothelial cells! It is clearly written in ref. 69 that they are dealing with SOD3 and not MnSOD!!! And so on… As there are 215 references, I did not check them entirely one by one. But with the numerous mistakes reported above, and even though these errors can be easily corrected (this has to be noted), one might wonder if what is written in this review can be fully trusted.   MINOR POINTS:   - The authors sometimes make what might be called an “error of precision” when describing the results of an article. An example is about ref. 60 in which B. longum has been engineered to deliver MnSOD in a rat model of colitis. In the article, one would think that B. longum naturally delivers a fusion protein containing hMnSOD given the way this passage is written.   - Many bibliographical references do not contain the page (and volume) numbers.   - The authors use many abbreviations without even defining them before use, e.g. in Figure 1 or in Table 1.   - Title of Figure 2 is misleading. In the same figure legend, the significance of the arrows must be suppressed and the meaning of ETC, Ac (…) must be clearly explained.   - A scheme detailing the structure of the different SOD mimics and compounds described in the review would have been welcome for bio-inorganic and medicinal chemists.

Author Response

  1. This referee thinks that making a more or less descriptive catalog of all the pathologies in which MnSOD appears is not read willingly and sometimes does not add anything more to what is already present in the literature. Thus, a restructuring of this review with a new introduction (ROS, ROS signaling, MnSOD – see below), a summary of the studies carried out (more or less described in the table) followed by an answer on emerging research questions on the analyzed topic and a conclusion of what can be developed and where we can hope to go next (roughly written is the last paragraph) would be preferable to a descriptive catalog. As presented, this referee feels that this review does not provide a clear and comprehensive overview of scientific advances in this field.
    Response: Thank you for the suggestion. We have accordingly modified the manuscript with a restructured new introduction and added the limitations and challenges for MnSOD-related studies in the discussion. Further prospects and development of the antioxidant therapy were also discussed to give a more comprehensive and clearer overview in the revised manuscript.
  2. The article would have clearly gained in clarity and impact if this review contained a clear and concise introductory paragraph on Reactive Oxygen Species (ROS), their definition, their genesis, their signaling functions under normal conditions and the involvement (among others) of MnSOD in these signaling functions. Indeed, it is now well-known that mitochondria drive +/- cellular functions by modulating the production of metabolites intervening as signaling molecules. mtH2O2 is essential for energetic/metabolic homeostasis and H2O2 drives cellular metabolism under physiological conditions. Thus, the question of how H2O2 is produced and regulated in the mitochondria is crucial. And H2O2 formation in mitochondria depends on and is controlled by MnSOD.
    Response: Agreed. We have added the relevant concepts, genesis and signalling pathways of ROS in the first paragraph of the introduction. The production and regulation of H2O2 were added in the second paragraph of the introduction.
  3. In line with the previous comment, the authors state that “MnSOD is a principal antioxidant enzyme that scavenges excessive ROS” in the abstract. This is a serious error. ROS include but are not exclusively superoxide radical anions! Furthermore, MnSOD converts O2°- into H2O2 AND O2.
    Response: Thank you for pointing this out. We have revised the description in the abstract.
  4. On the other hand, a more thorough paragraph on MnSOD must be presented in this review. It could deal with its biogenesis, its metallation process then assembly into its quaternary structure (see for instance articles from Valeria Culotta) and its regulation (acetylation at K68, phosphorylation at Ser106, regulation via the mitochondrial protein influx...) as well as the place and the effects of this regulation on the properties catalytic activity, stability...) and functions of the enzyme. In addition, when describing metallation, a brief allusion to the replacement of Mn2+ by Fe2+ in cases of Mn2+ deficiency could be made, this leading to a pro-oxidant function of the enzyme via a peroxidase activity.
    Response: As suggested by the reviewer, we have added the biogenesis, metalation process, quaternary structure and regulation of MnSOD in the third paragraph of the introduction.
  5. The above paragraph could be briefly completed by the various signals that leads to MnSOD down-regulation or up-regulation (instead of its overexpression thanks to lenlentivirus or transgene) as well as by the various modifications/inhibitions of MnSOD observed in (pathological) conditions such as nitration by peroxynitrite, product-inhibited complex… This would make it possible to establish a better connection with all the pathologies addressed by the authors in which MnSOD is involved.
    Response: To improve the link between MnSOD and pathologies, we have added the regulation of MnSOD at multiple levels in the third paragraph of the introduction.
  6. According to this referee, the authors make some small “shortcuts”, approximations, in the introductory paragraph and the abstract. For example, the authors write that "these observations indicate that MnSOD is widely involved in multiple biological processes...". Nevertheless, is it really MnSOD or variations in the concentrations of its substrate (O2°-) or products (in particular H2O2 and variation of its signaling)?
    Response: To clarify this issue, we have amended the sentences in the abstract and introduction. We also discussed the relevant questions in the second paragraph of the discussion.
  7. Also, “The SOD family is in the front line of antioxidant enzymes that defend against ROS”. And yet, SOD produces ROS (H2O2) whose high concentrations are pathogenic... Hence, the necessity of a more thorough introductory paragraph on ROS, MnSOD and the other enzymes that are the first bastion of defense against ROS (catalase (CAT) and glutathione peroxidase (GPx)).
    Response: We have included the relevant description of ROS, MnSOD and other antioxidant enzymes in the first and the second paragraph of the introduction.
  8. In addition, what are “ROS-related diseases” (abstract)? Are ROS the source of the diseases, the consequence…? It reads as if ROS were responsible for the listed diseases whereas ROS are mediators of cellular damage in the described diseases including for example diabetes in which acute or chronic high glucose increases the production of ROS.
    Response: We have clarified the sentences in the abstract. We also discussed the relevant questions in the second paragraph of the discussion.
  9. Finally, the article contains numerous errors regarding references that are not properly adequate or are misused. For instance, in the introductory paragraph, ref. 1 deals specifically with mtROS and mtROS signaling and is thus inappropriate to describe the dynamic balance between ROS production/degradation and the maintenance of cellular redox homeostasis. Ref. 10 discusses the use of MnSOD and SOD mimics as radioprotectors and is therefore unsuitable to describe its strategic location and its role in superoxide dismutation during cellular respiration and its function in the maintenance of redox homeostasis. Ref. 11 deals with increased acetylated levels of MnSODK68 in patients harboring IPF. Accordingly, MnSOD gene expression is not downregulated in this case. Similarly, ref. 13 debates about MnSOD as a target of DAQ with a resulting 50% loss in its enzymatic activity. No downregulation in here too. Unfortunately, this point (references that are not truly adequate or misplaced) is recurrent throughout the article… For example, paragraph 2.1.1 begins with ref. 54 that only and solely discusses the treatment of a rat model of lung pleurisy with the MnSOD mimetics MnTBAP! Hence, only one model of lung damage and not “different modelS” as stated by the authors. However, refs. 17,18,54 and 55 (altogether) showed indeed that “MnSOD was found to attenuate pulmonary inflammation in different models of lung damage”. Then a very serious error is found in paragraph 2.1.4 when the authors state that “the elevation of MnSOD activity was also observed in plasma” (ref. 69). But the major SOD isoenzyme in plasma is extracellular SOD (EC-SOD) that is mostly secreted by endothelial cells! It is clearly written in ref. 69 that they are dealing with SOD3 and not MnSOD!!! And so on… As there are 215 references, I did not check them entirely one by one. But with the numerous mistakes reported above, and even though these errors can be easily corrected (this has to be noted), one might wonder if what is written in this review can be fully trusted.
    Response: We sincerely appreciate the reviewer’s careful reading. We have corrected or replaced these improper or misused references and relevant descriptions in the main text. We have also checked all references throughout the manuscript and modified some errors.
  10. The authors sometimes make what might be called an “error of precision” when describing the results of an article. An example is about ref. 60 in which B. longum has been engineered to deliver MnSOD in a rat model of colitis. In the article, one would think that B. longum naturally delivers a fusion protein containing hMnSOD given the way this passage is written.
    Response: We have revised the sentence to make it clearer, and checked the problems throughout the manuscript.
  11. Many bibliographical references do not contain the page (and volume) numbers.
    Response: We checked and corrected all the references format in the revised manuscript.
  12. The authors use many abbreviations without even defining them before use, e.g. in Figure 1 or in Table 1.
    Response: All abbreviations used in the figures and table were added in the figure legends or table footer.
  13. Title of Figure 2 is misleading. In the same figure legend, the significance of the arrows must be suppressed and the meaning of ETC, Ac (…) must be clearly explained.
    Response: We have changed the tile of Figure 2, and corrected the arrow ‘⊥’ to represent suppressed. The abbreviations of ETC and Ac were explained in the figure legend.
  14. A scheme detailing the structure of the different SOD mimics and compounds described in the review would have been welcome for bio-inorganic and medicinal chemists.
    Response: Thanks for the suggestion. We have added the available chemical structures in Table 1.

Reviewer 3 Report

I have reviewed the manuscript IJMS-2044152. This review article organizes the mechanisms of Manganese superoxide dismutase (MnSOD) in ROS-related diseases, which inculding in vivo or in vitro studies of fibrotic diseases, inflammation, diabetes, vascular diseases, neurodegenerative diseases, and cancer. The article is complete and detailed, which could help develop a strategy in the clinic and understand the therapeutic role of MnSOD activators and MnSOD mimetics; it also provides essential insight into the prevention and treatment of several diseases.

Author Response

We sincerely appreciate your recognition of our work.

Round 2

Reviewer 1 Report

The manuscript can be accepted this form.

Author Response

(The authors gave the same response as above.)

Reviewer 2 Report

Mengfan Liu et al present here a revised version of their article on MnSOD and its role in several diseases. I did previously raise several points encountered during the reading of this article. Consequently, the authors furnished a point-by-point answer to the comments made, edited their manuscript (mainly the abstract, the introduction, the conclusion and the proper use of the references listed in the article), and somewhat arranged the organizing of their manuscript by providing a more complete introductory paragraph and by adding a paragraph on the limitations of the MnSOD studies in their concluding paragraph. I would like to thank them for that despite the fact that the core of the manuscript has not been restructured. It may be too difficult in a short time, or the authors think it's a waste of time to restructure the whole article so that it doesn't look like a hard-to-read descriptive catalog of all the pathologies in which MnSOD appears. It will be up to the scientific community to judge the thing. I thus believe that this article can be published in its revised version after few corrections, as follows:

1-  Abstract: Overproduction of reactive oxygen species (ROS) disrupts the body's antioxidant defence, compromising redox homeostasis and oxidative stress...”. Or (?) “Overproduction of reactive oxygen species (ROS) disrupts the body's antioxidant defence, compromising redox homeostasis and INCREASING oxidative stress...”

2-  Abstract: “and understanding the therapeutic role of MnSOD provides positive insight into preventing and treating related diseases”. Please add MAY in the sentence, that is “and understanding the therapeutic role of MnSOD MAY provide positive insight into preventing and treating related diseases”

Please be careful about the new introductory paragraph. There are too many English mistakes. Please do correct them! Sometimes, even the verb is missing in the sentence, e.g. “MnSOD, a nuclear-encoded enzyme that is translocated into the mitochondrial matrix[12]”... Also,

3-  Introduction: “H2O2 is (…) directly generated by enzymes such as NOX4, monoamine oxidases and xanthine oxidases”. What about superoxide dismutase (SOD)???

4-  Introduction: “ROS is required for basal cellular processes”. Which one? Please precise.

5-  Introduction: I do not understand this sentence “While to scavenge O2•, catalyzing the dismutation of O2•- to H2O2 and O2 by SOD family is the most effective reaction”. Please re-write

Author Response

  1. Abstract: “Overproduction of reactive oxygen species (ROS) disrupts the body's antioxidant defence, compromising redox homeostasis and oxidative stress...”. Or (?) “Overproduction of reactive oxygen species (ROS) disrupts the body's antioxidant defence, compromising redox homeostasis and INCREASING oxidative stress...”
    Response: Agreed. We have added “increasing” in the sentence.
  2. Abstract: “and understanding the therapeutic role of MnSOD provides positive insight into preventing and treating related diseases”. Please add MAY in the sentence, that is “and understanding the therapeutic role of MnSOD MAY provide positive insight into preventing and treating related diseases”
    Response: We have added “may” in the sentence.
  3. Please be careful about the new introductory paragraph. There are too many English mistakes. Please do correct them! Sometimes, even the verb is missing in the sentence, e.g. “MnSOD, a nuclear-encoded enzyme that is translocated into the mitochondrial matrix[12]”.
    Response: We have revised the above sentence as suggested, and improved the English language of the introduction part in the revised manuscript.
  4. Introduction: “H2O2 is (…) directly generated by enzymes such as NOX4, monoamine oxidases and xanthine oxidases”. What about superoxide dismutase (SOD)?
    Response: Thank you for pointing this out. We have updated the SOD in the sentence.
  5. Introduction: “ROS is required for basal cellular processes”. Which one? Please precise.
    Response: To clarify this issue, we have modified the description.
  6. Introduction: I do not understand this sentence “While to scavenge O2•, catalyzing the dismutation of O2•- to H2O2 and O2 by SOD family is the most effective reaction”. Please re-write.
    Response: We have revised the sentence to make it clearer.
